# Impact of Fly Ash as a Raw Material on the Properties of Refractory Forsterite–Spinel Ceramics

**Martin Nguyen ***  **and Radomír Sokolář**

Faculty of Civil Engineering, Institute of Technology of Building Materials and Components,
Brno University of Technology, Veveří 331/95, 602 00 Brno, Czech Republic; sokolar.r@fce.vutbr.cz
* Correspondence: nguyen.m@fce.vutbr.cz

**Abstract:** This article examines the process for the synthesis of forsterite–spinel ($2MgO \cdot SiO_2/MgO \cdot Al_2O_3$) refractory ceramics from fly ash and alumina as sources of aluminum oxide. Raw materials were milled, mixed in different ratios and sintered at 1500 °C for 2 h. Sintered samples were characterized by XRD, thermal analyses and SEM. Porosity, water absorption, bulk density, refractoriness, refractoriness under load and thermal shock resistance were also investigated. The impact of fly ash as a raw material was investigated in accordance with the resulting properties and microstructure of samples with fly ash and alumina as the raw materials. Due to the positive effect of flux oxides (iron oxides and alkalis) on sintering, the mullite contained in fly ash completely decomposed into silica and alumina, which, together with magnesium oxide, formed spinel. This led to improved microstructural and mechanical properties and thermal shock resistance. In particular, mixtures with 10 wt.% and 20 wt.% of fly ash had the most promising results compared to alumina mixtures. Both modulus of rupture and thermal shock resistance were improved, while the impact on refractory properties was minimal. The novelty of this research lies in the recycling of fly ash, a by-product from coal-burning power plants, into a raw material for the production of forsterite–spinel refractory ceramics.

**Keywords:** sintering; spinel; forsterite; refractory ceramics; fly ash; alumina

## 1. Introduction

Refractory forsterite ceramics are an important type of material and are used in the metallurgical and cement industries as the lining of metallurgical furnaces and rotary kilns due to their high melting point of 1890 °C [1–3]. Forsterite is also used in electrotechnical engineering for ceramic–metal joints because of its high thermal expansion coefficient, which is similar to the coefficient of metals [4,5]. To date, various studies have been conducted on the use of forsterite as a crystal in ring cavity lasers [6], as a promising biomaterial for bone transplants [7,8] and as a material for composite nanomaterials [9–11].

As a result of its high thermal expansion coefficient, forsterite has a low thermal shock resistance. This can be improved by the addition of magnesium–alumina spinel (MA-spinel; commonly abbreviated as spinel), which improves the microstructure, the mechanical properties and the thermal shock resistance [12,13]. Spinel can be synthesized from aluminum and magnesium oxides. Generally, spinel is synthesized from alumina or bauxite and magnesium oxide [9,14,15]. However, fly ash, which is obtained as a secondary product from coal-burning power plants, can be used as an inexpensive source of aluminum and silicon oxides, which in turn can be used as raw materials for sintering refractory ceramics [16–21]. Despite the utilization of fly ash in the synthesis of aluminosilicate refractories, little research has been carried out on the use of fly ash in the synthesis of other refractories containing silicon and/or aluminum oxides, such as forsterite–spinel ceramics. The mullite-containing

fly ash decomposes into silica and alumina in the presence of flux oxides, and these, together with magnesium oxide, form spinel, as proven by both the literature [17–19] and previous research [22,23]. Magnesium–alumina spinel is used with magnesia as a lining in cement and lime kilns because of its high melting point of 2135 °C, low thermal expansion coefficient in comparison with forsterite and good thermal shock, chemical and corrosion resistance [9,12,13].

The main objective of this work was to determine the impact of fly ash as a raw material on the synthesis of refractory forsterite–spinel ceramics. The novelty of this work lies in the recycling of fly ash, a waste product of coal-burning power plants, into a raw material for the production of refractory forsterite–spinel ceramics, while comparing the resulting properties with the synthesis of this refractory ceramic from alumina as a raw material. Physico-mechanical properties, phase composition, microstructure, thermal analyses and refractory and mechanical properties were investigated.

## 2. Materials and Methods

### 2.1. Raw Materials

Raw materials were obtained from multiple sources. Olivine from the Norwegian company A/S Olivin, calcined caustic magnesite (CCM 85) from SMZ Jelšava (Slovakia), coal fly ash class F (according to standard ASTM C618; $d_{50}$ = 14 μm) from power plant Mělník (Czech Republic), reactive alumina CTC 22 ($d_{50}$ = 1.9 μm) from Almatis (Germany) and kaolin Sedlec Ia ($d_{50}$ = 1.3 μm) from Sedlecký kaolin (Czech Republic). Table 1 presents the chemical composition of the used materials, which was determined by chemical composition analysis and X-ray fluorescence (XRF).

**Table 1.** The chemical composition of used raw materials.

| Raw Materials | MgO [%] | SiO$_2$ [%] | Al$_2$O$_3$ [%] | CaO [%] | Fe$_2$O$_3$ [%] | K$_2$O + Na$_2$O [%] | LOI * [%] |
|---|---|---|---|---|---|---|---|
| CCM 85 | 85.0 | 0.5 | 0.8 | 5.2 | 7.30 | 0.20 | 1.0 |
| Olivine | 24.1 | 64.7 | 1.0 | 0.7 | 8.80 | 0.50 | 1.0 |
| Fly ash | 1.4 | 57.3 | 29.3 | 2.2 | 5.10 | 1.70 | 1.2 |
| Alumina | 0.0 | 0.0 | 99.7 | 0.0 | 0.03 | 0.12 | 0.1 |
| Kaolin | 0.5 | 46.8 | 36.6 | 0.7 | 0.85 | 1.20 | 13.2 |

* Loss on ignition.

A total of eight different mixtures were designed and divided into two sets, depending on the source of aluminum oxide (Al$_2$O$_3$) for the synthesis of spinel. The first set of four mixtures had fly ash (FA10-FA40) as the source of Al$_2$O$_3$, and the second set of four mixtures had reactive alumina (RA10-RA40) as the source of Al$_2$O$_3$. The quantity of fly ash in the mixtures chosen ranged from 10–40 wt.% with 10 wt.% increments and the quantity of alumina was calculated to match the exact content of Al$_2$O$_3$ present in the fly ash mixtures. The designed compositions of all the mixtures used to obtain stoichiometric forsterite are presented in Table 2.

**Table 2.** Designed compositions of all raw material mixtures.

| Mixtures | CCM 85 [wt.%] | Olivine [wt.%] | Fly Ash Mělník [wt.%] | Alumina CTC 22 [wt.%] | Kaolin Sedlec Ia [wt.%] |
|---|---|---|---|---|---|
| FA10 | 43.2 | 41.8 | 10.0 | - | 5.0 |
| RA10 | 41.0 | 51.1 | - | 2.9 | 5.0 |
| FA20 | 44.1 | 30.9 | 20.0 | - | 5.0 |
| RA20 | 39.8 | 49.4 | - | 5.8 | 5.0 |
| FA30 | 45.0 | 20.0 | 30.0 | - | 5.0 |
| RA30 | 38.6 | 47.8 | - | 8.6 | 5.0 |
| FA40 | 45.9 | 9.1 | 40.0 | - | 5.0 |
| RA40 | 37.4 | 46.1 | - | 11.5 | 5.0 |

The ternary phase diagram of the MgO-Al$_2$O$_3$-SiO$_2$ system is presented in Figure 1 with all designed mixtures plotted on the line between forsterite and spinel.

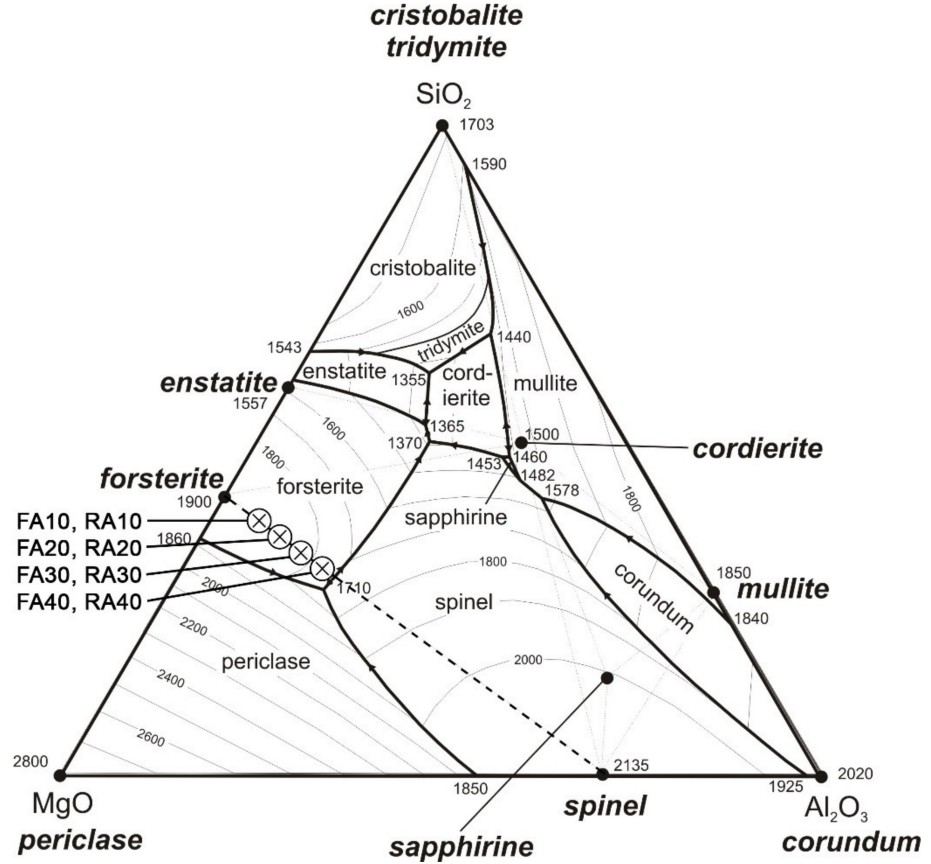

**Figure 1.** Ternary phase diagram of MgO-Al$_2$O$_3$-SiO$_2$ system with plotted mixtures.

Figure 2A–C represent the scanning electron microscope (SEM, Tescan Mira 3, Tescan Orsay Holding a.s., Brno, Czech Republic) microphotographs of untreated olivine, alumina and fly ash. There is an apparent fibrous microstructure to untreated olivine. Alumina was finely ground to the desired particle size, where d$_{50}$ = 1.9 μm. Fly ash has its typical spherical particles with a diameter between 0.4–90 μm (d$_{50}$ = 14 μm).

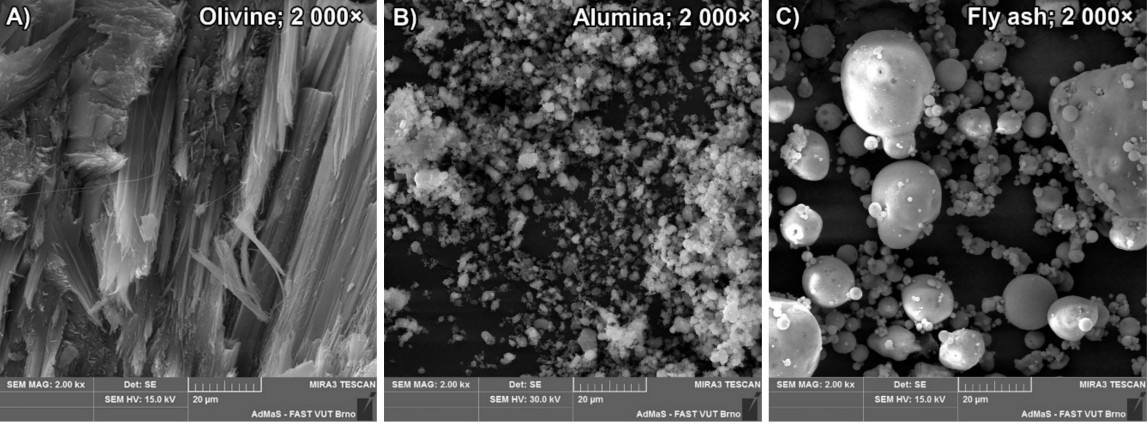

**Figure 2.** SEM microphotographs of selected raw materials: (**A**) olivine, (**B**) alumina and (**C**) fly ash; magnification = 2000×.

The mineralogical composition of the raw materials is presented in Figure 3. CCM 85 was primarily composed of periclase with trace quantities of iron compounds. The major crystal phase in olivine was forsterite ($2MgO \cdot SiO_2$) with minor phases of fayalite ($2FeO \cdot SiO_2$), serpentinite ($3MgO \cdot 2SiO_2 \cdot 2H_2O$) and quartz ($SiO_2$). The Class-F fly ash was primarily composed of mullite ($3Al_2O_3 \cdot 2SiO_2$) and quartz ($SiO_2$). Both olivine and fly ash had background curvature, indicating the presence of an amorphous glass phase. Reactive alumina was composed of almost pure corundum ($Al_2O_3$), and kaolin was primarily kaolinite ($Al_2O_3 \cdot 2SiO_2 \cdot 2H_2O$) with traces of biotite ($K(Mg,Fe)_3AlSi_3O_{10}(F,OH)_2$).

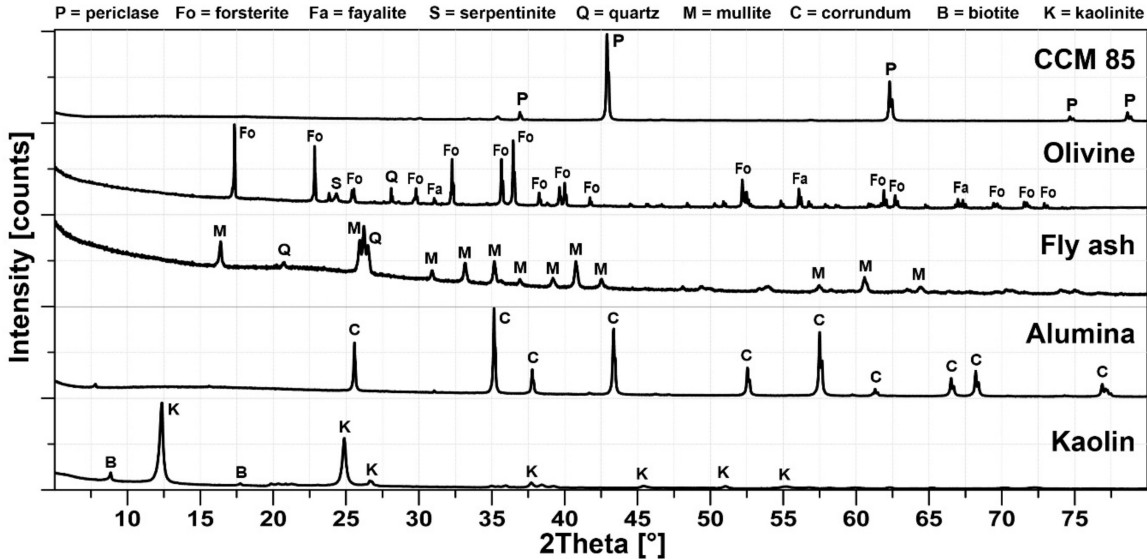

**Figure 3.** X-ray diffraction analysis of the used raw materials.

*2.2. Experiment*

All mixtures were prepared by milling olivine and CCM to a particle size that ranged between 1 and 80 μm (where $d_{50} = 10$–$20$ μm) and then mixing with other fine raw materials. Particle size distribution was determined by laser granulometry (Malvern Mastersizer, Malvern Panalytical, Malvern, United Kingdom). Mixtures were then homogenized in a rotary mechanical homogenizer for 24 h, then mixed with water to gain the optimal plasticity using Pfefferkorn apparatus (standard ČSN 72 1074). Samples were then molded into brass molds from plastic paste. The approximate dimensions of the test samples were $20 \times 25 \times 100$ mm$^3$ prisms, cylinders with a diameter of 50 mm and a height of 50 mm for refractoriness under load and $230 \times 64 \times 54$ mm$^3$ prisms (dimensions of half of a standard brick) for thermal shock resistance. The pyrometric cones used for the refractoriness experiments (standard EN 993-13:1995) were also prepared from the same mixtures in a set of three for each mixture. All test samples were then dried in a laboratory dryer at 105 °C. After drying, test samples were fired in a laboratory furnace with an air atmosphere at 1500 °C, which used a heating rate of 4 K/min and a soaking time of 2 h at maximum temperature.

Test samples were then subjected to several experimental procedures that assessed the apparent porosity, water absorption and bulk density, which were determined by a vacuum water absorption method with subsequent hydrostatic weighing (standard EN 993-1:1995); thermal analyses: thermogravimetry (TG), derivative thermogravimetry (DTG; Mettler Toledo TGA/DSC1, standard ČSN 72 1083); and change in dimension during firing (standard EN 993-10:1997). Refractoriness of pyrometric cones (standard EN 993-12:1997) was performed with a set of three identical pyrometric cones from each mixture in a laboratory furnace with an observation port with a camera which enabled real-time observation in the furnace and allowed for the capture of the exact moment of the bending of pyrometric cones. Refractoriness under load (standard ISO 1893:2007) was performed on cylindrical samples according to the standard and the temperature at 0.5% deformation ($T_{0.5}$) was measured.

Determination of thermal shock resistance was carried out according to standard EN 993-11:2007 method B, which defines parameter "residual modulus of rupture (MOR)", this parameter enables a quantitative approach for measuring thermal shock resistance. It is a ratio between the modulus of rupture (MOR) of cycled samples and the MOR for samples at a normal laboratory temperature of 25 °C. Cycling of samples was done according to the standard EN 993-11:2007. Thermal dilatometric analysis (standard EN 993-19:2004); modulus of rupture (MOR; Testometric M350-20CT, standard EN 993-6:1995); and X-ray diffraction analysis (XRD; Panalytical Empyrean, PANalytical B.V., Almelo, Netherlands) that used CuK$\alpha$ as a radiation source, an accelerating voltage of 45 kV and a beam current of 40 mA and SEM with an EDX probe (Tescan Mira 3, Tescan Orsay Holding a.s., Brno, Czech Republic) were used to determine morphology and elemental analysis of the crystal structure.

## 3. Results and Discussion

The main aim of this paper was to investigate the impact of fly ash as the raw material used during production on the resulting properties of refractory forsterite–spinel ceramics. A comparison of mixtures that used fly ash with mixtures that used alumina with equal proportions of aluminum oxide was investigated.

### 3.1. Mineralogical Composition and Microstructure

The X-ray diffraction (XRD) analysis of samples of all the designed mixtures with an equal proportion of $Al_2O_3$ content is presented in Figure 4. All mixtures contained forsterite ($2MgO \cdot SiO_2$), spinel ($MgO \cdot Al_2O_3$), periclase (MgO) and monticellite ($CaO \cdot MgO \cdot SiO_2$) minerals. The curved background of the XRD diffractogram also indicated the presence of an amorphous glass phase, and the background noise indicated the presence of iron oxides ($Fe_2O_3$—hematite, $Fe_3O_4$—magnetite) due to the use of CuK$\alpha$ as a radiation source.

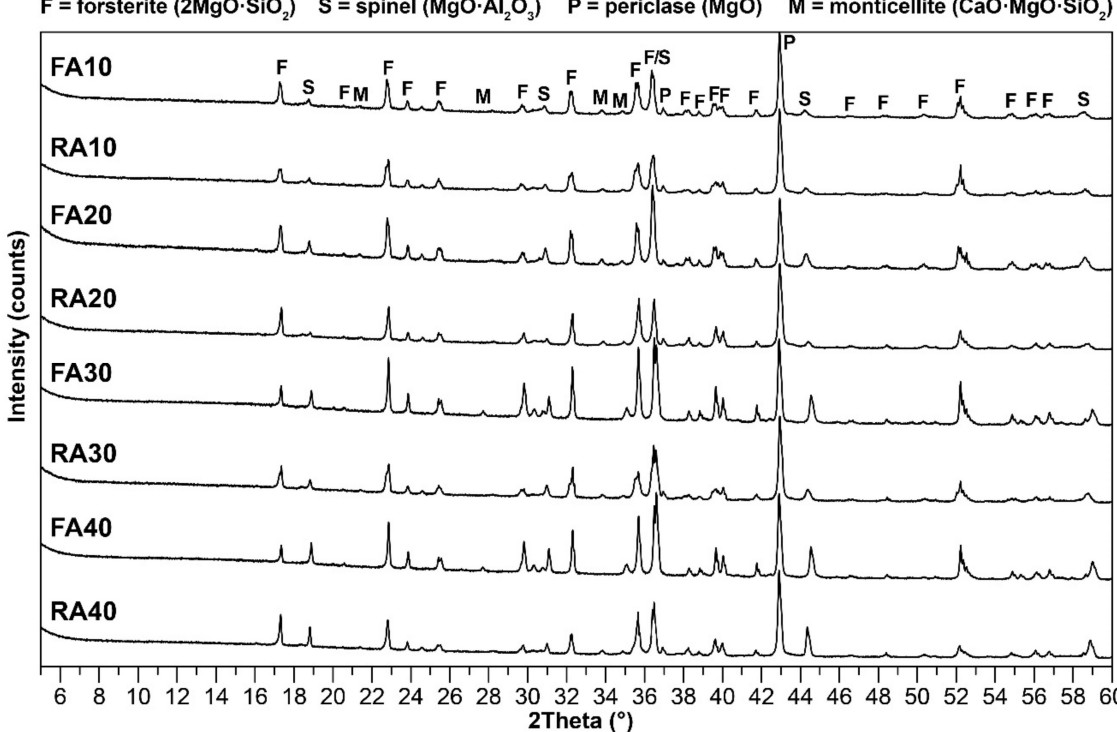

**Figure 4.** Comparison of the X-ray diffraction analysis of all designed samples with fly ash (FA10-FA40) or alumina (RA10-RA40).

As can be seen in Figure 3, XRD analysis confirmed the presence of mullite in fly ash. However, no traces of mullite were found in the XRD of fired samples which used fly ash (Figure 4). At the same time, XRD analysis confirmed the presence of corundum in alumina (Figure 3), but no traces of corundum were found in the XRD of the fired samples which used alumina. It can therefore be concluded that all mullite from fly ash, and all corundum from alumina, transformed into spinel.

Similarly, as can be seen from Figure 3, olivine contained minor quantities of fayalite ($2FeO \cdot SiO_2$), which decomposed at 1205 °C into iron(II) oxide and silica. Iron(II) oxide then oxidized into hematite ($Fe_2O_3$) and magnetite ($Fe_3O_4$), and silica reacted with magnesium oxide into forsterite [24]. The decomposition of fayalite can also be observed in the thermal analyses (Figures 6 and 7) as an endothermic peak above 1200 °C. The presence of iron oxides was indicated in the XRD patterns (Figure 3) as background noise, which was more prominent in samples using fly ash. Trace quantities of monticellite ($CaO \cdot MgO \cdot SiO_2$) were also present in all samples due to the calcium oxide content, mainly in CCM (5.2%) and fly ash (2.2%). The presence of unreacted magnesium oxide (periclase) can be explained by the fact that part of the silicon dioxide transformed due to the enhanced sintering effect of flux oxides into an amorphous glass phase, which is indicated in Figure 4 as a curved background in the XRD diffractogram.

Morphology and microstructure were examined by SEM with energy-dispersive X-ray spectroscopy (EDX), microphotographs are presented in Figure 5 A–D. Spinel crystals formed in patches in both fly ash and alumina mixtures with a diameter range of 2–4 μm. Spinel crystals were located on the edges of the larger forsterite matrix or connected forsterite patches together. Spinel crystals that formed from the alumina mixtures were uniform with a smooth surface and no cracks, whereas spinel crystals that formed from fly ash mixtures were cracked with indented and irregular edges. This can be explained by the fact that spinel crystals from alumina mixtures formed directly from magnesium oxide and reactive aluminum oxide, whereas spinel crystals from fly ash mixtures formed indirectly from mullite decomposition into aluminum oxide and silicon dioxide and subsequent reaction between aluminum oxide with magnesium oxide.

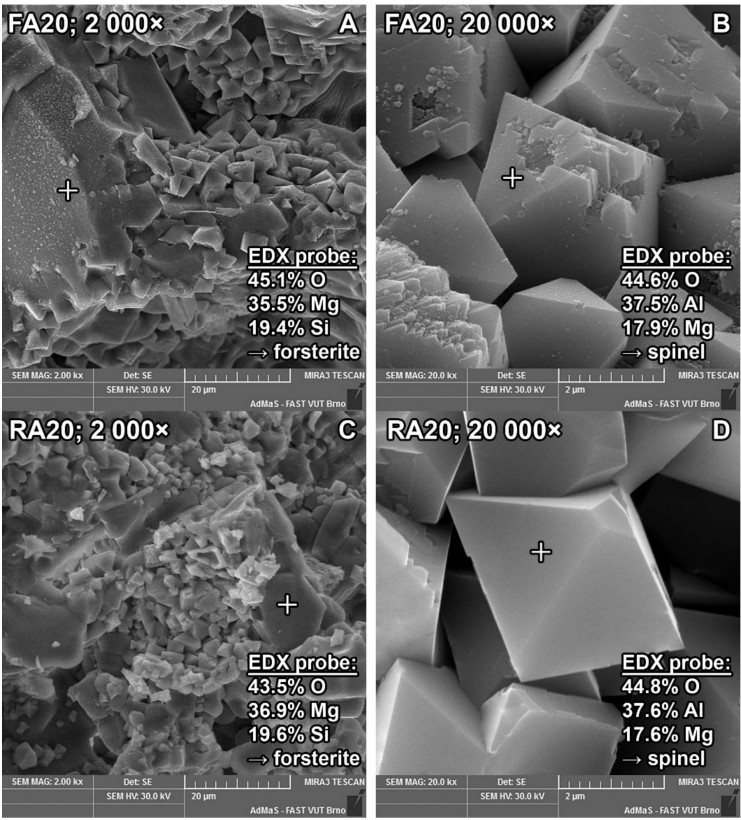

**Figure 5.** SEM images of mixtures (**A**,**B**) FA20 and (**C**,**D**) RA20 with magnifications of 2000× (left) and 20,000× (right) with EDX probe results.

### 3.2. Thermal and Thermo-Mechanical Analyses

Thermal analyses (Figures 6 and 7) for both fly ash and alumina mixtures were almost identical until temperatures reached 1200 °C and above. Endothermic peaks on the DTG curve between 100 and 200 °C correspond to the loss of physically bound water, while peaks between 350–380 °C and 430–480 °C correspond to the dehydroxylation of serpentinite (Equation (1)) [24].

$$3MgO \cdot 2SiO_2 \cdot 2H_2O \rightarrow 3MgO \cdot 2SiO_2 + 2H_2O. \tag{1}$$

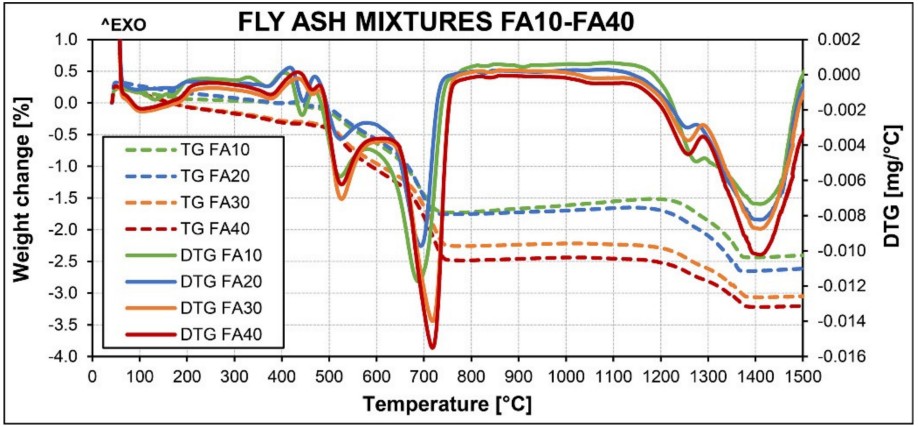

**Figure 6.** Thermal analyses of mixtures with fly ash (FA10-FA40).

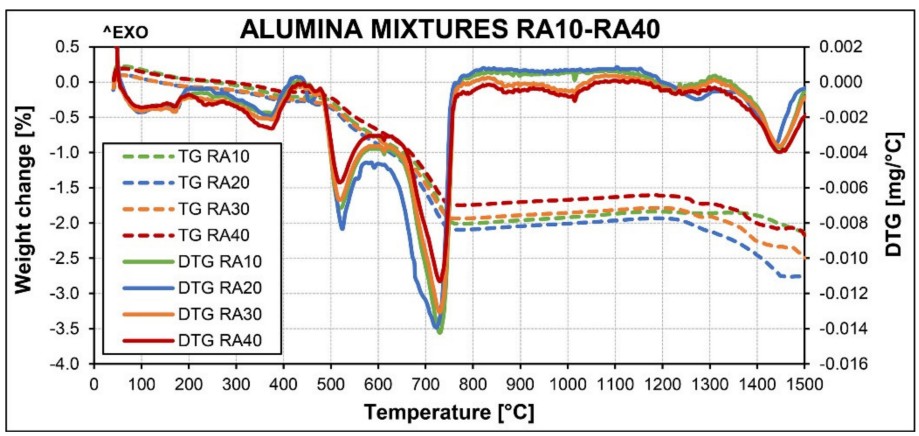

**Figure 7.** Thermal analyses of mixtures with reactive alumina (RA10-RA40).

Peaks around 530 °C correspond to a modification change from α-quartz to β-quartz, and peaks around 720 °C correspond to the dehydroxylation of kaolinite to form metakaolin (Equation (2)):

$$Al_2O_3 \cdot 2SiO_2 \cdot 2H_2O \rightarrow Al_2O_3 \cdot 2SiO_2 + 2H_2O. \tag{2}$$

Endothermic peaks in fly ash mixtures (Figure 6) at 1260 °C and 1400 °C correspond to fayalite (Equation (3)) and mullite (Equation (4)) decomposition:

$$2FeO \cdot SiO_2 \rightarrow 2FeO + 2SiO_2, \tag{3}$$

$$3Al_2O_3 \cdot 2SiO_2 \rightarrow 3Al_2O_3 + 2SiO_2, \tag{4}$$

with subsequent spinel (Equation (5)) and forsterite (Equation (6)) formation from the decomposition products, along with the creation of the liquid glass phase [17,21]:

$$MgO + Al_2O_3 \rightarrow MgO \cdot Al_2O_3, \tag{5}$$

$$2MgO + SiO_2 \rightarrow 2MgO \cdot SiO_2. \tag{6}$$

Broad endothermic peaks in alumina mixtures (Figure 7) at 1230–1290 °C also correspond to fayalite decomposition, and the peaks at 1450 °C correspond to spinel creation from magnesium and aluminum oxides [17,21,24].

Thermal dilatometric analysis of dried unfired samples is presented in Figure 8. For better clarity and demonstration, only four samples were selected: FA10 and RA10, which respectively had the lowest quantities of fly ash and alumina; and FA40 and RA40, which respectively had the highest quantities of fly ash and alumina. Noticeable differences between mixtures with fly ash and alumina were observed above 1000 °C. Samples with fly ash had four different stages. In the first stage, there was a linear expansion of raw materials that occurred up to 1000 °C, which was then followed by a shrinkage stage in the temperature interval of 1000–1250 °C. The third stage observed an expansion in the temperature interval 1250–1350 °C, which was caused by mullite crystal growth and reaction with magnesium oxide to form spinel, leading to the creation of the liquid glass phase. This behavior was also reported in other works [17,21] and previous research [22,23]. The fourth stage occurred at temperatures above 1350 °C and caused re-shrinkage due to the sintering promotion action of the liquid glass phase [17,21].

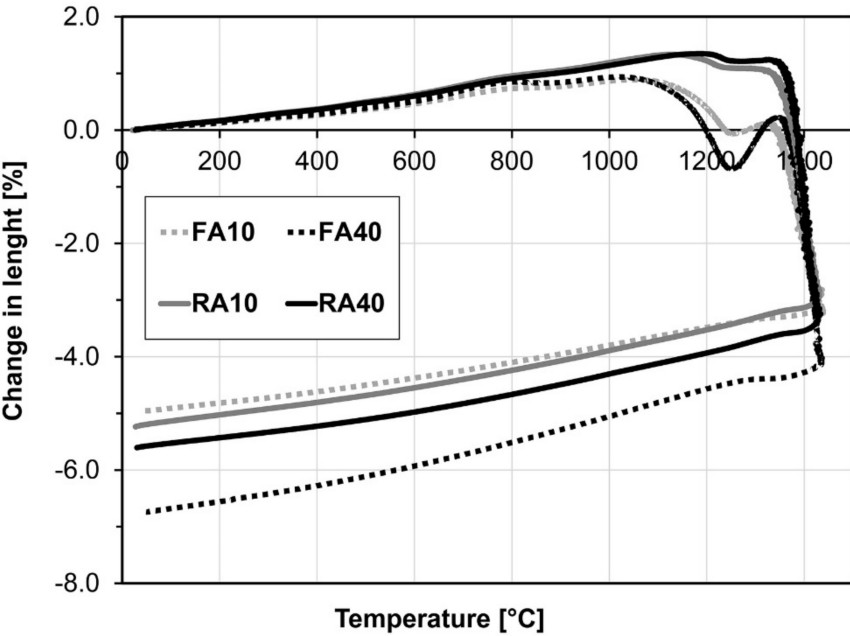

**Figure 8.** Selected results of the thermal dilatometric analysis.

Results of refractoriness under load (Figures 9 and 10) showed that in mixtures with increasing amounts of fly ash, the thermal expansion is slightly higher due to the higher content of the amorphous glass phase. With increasing amounts of fly ash in the mixture, the temperature at 0.5% deformation decreased due to the effect of flux oxides and the amorphous glass phase. Differential curves of mixtures with alumina were almost identical, including temperatures at 0.5% deformation. The higher thermal expansion in alumina mixtures than in fly ash mixtures can be explained by the higher amount of unreacted periclase (MgO) which has a higher coefficient of thermal expansion than forsterite or spinel.

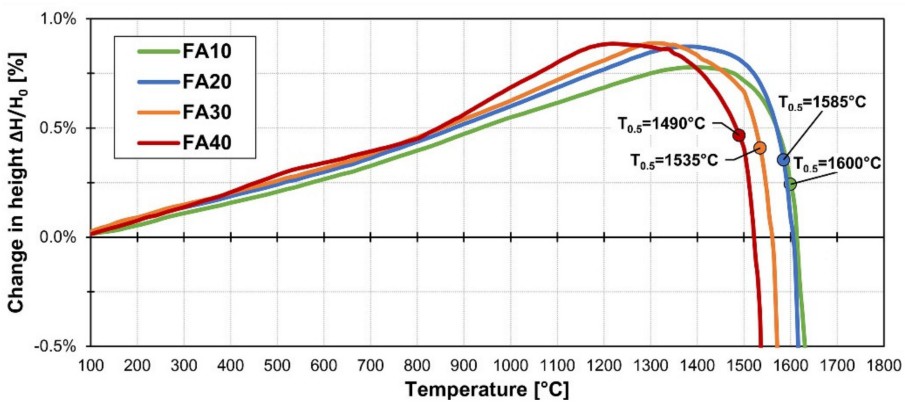

**Figure 9.** Results of refractoriness under load of mixtures with fly ash.

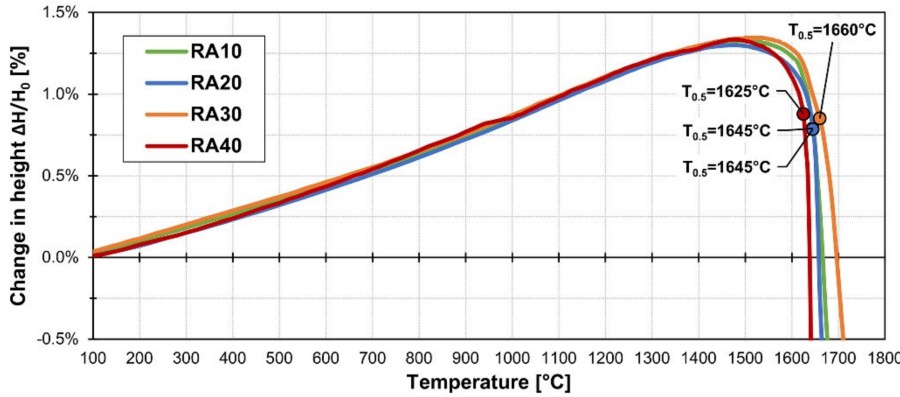

**Figure 10.** Results of refractoriness under load of mixtures with alumina.

### 3.3. Physico-Mechanical Parameters

As presented in Table 3, with increased quantities of fly ash in the mixture, the firing shrinkage also increased due to the higher quantities of flux oxides, such as iron oxide and alkalis. Higher quantities of flux oxides improve sintering and solidify the structure, leading to reduced porosity and water absorption while increasing bulk density and MOR. However, flux oxides have a negative impact on refractory properties, reducing both refractoriness and refractoriness under load. The refractoriness was reduced by 6% and the refractoriness under load $T_{0.5}$ was reduced by 7% between samples that had between 10 and 40 wt.% of fly ash in the mixture. With increasing amounts of alumina in the mixture, the MOR of these samples increased from 12.8 MPa (RA10) to 22.4 MPa (RA40). With increasing amounts of fly ash in the mixture, the MOR increased up to 30 wt.% of the fly ash. The FA30 mixture had the highest MOR, at 19.5 MPa, followed by the FA20 mixture, with a MOR at 17.4 MPa. Increased amounts of spinel in the mixtures lead to better microstructure and MOR due to the presence of spinel crystals on the edges of larger forsterite crystal patches, connecting them together, or interlocking spinel crystals embedded in amorphous glass matrix. Spinel ceramics also have a higher MOR and compressive strength then forsterite ceramics [1,2].

**Table 3.** The results of the aforementioned experiments for all mixtures.

| Mixtures | FA10 | RA10 | FA20 | RA20 | FA30 | RA30 | FA40 | RA40 |
|---|---|---|---|---|---|---|---|---|
| Firing shrinkage [%] | 7.2 | 7.7 | 7.5 | 8.6 | 11.5 | 10.3 | 13.7 | 11.5 |
| Apparent porosity [%] | 31.1 | 22.8 | 35.2 | 20.6 | 20.3 | 18.6 | 12.7 | 16.8 |
| Water absorption [%] | 13.3 | 8.5 | 16.1 | 7.6 | 9.4 | 6.7 | 4.5 | 6.0 |
| Bulk density [kg·m$^{-3}$] | 2340 | 2670 | 2190 | 2720 | 2230 | 2760 | 2390 | 2810 |
| Refractoriness [°C] | 1695 | 1730 | 1670 | 1715 | 1645 | 1700 | 1595 | 1690 |
| Refractoriness under load $T_{0.5}$ [°C] | 1600 | 1645 | 1585 | 1645 | 1535 | 1660 | 1490 | 1625 |
| Modulus of rupture (MOR) [MPa] | 13.5 | 12.8 | 17.4 | 19.2 | 19.5 | 20.7 | 15.1 | 22.4 |
| Residual MOR [%] | 5.9 | 20.3 | 16.7 | 19.8 | 21.9 | 26.9 | 9.4 | 14.2 |

Increasing the amounts of aluminum oxide in the mixtures led to improved microstructure and mechanical properties, particularly MOR and thermal shock resistance. However, the mixture FA40, which had 40 wt.% of fly ash, contained excessive levels of flux oxides that led to the creation of greater amounts of the amorphous glass phase (see more prominent expansion stage at 1250 °C in Figure 8), which caused lower mechanical and refractory properties than other mixtures using fly ash. Similarly,

the mixture with 30 wt.% of fly ash had a significant decrease in temperature in refractoriness under load (Figure 9) due to the higher levels of the amorphous glass phase.

## 4. Conclusions

Refractory forsterite–spinel ceramics were sintered from mixtures with fly ash and alumina to compare the resulting properties and effects of flux oxides (iron oxides and alkalis) contained in the raw materials. The spinel crystals formed from alumina and magnesium oxide were more uniform and without cracks, and the resulting properties, such as MOR and thermal shock resistance, improved with increased alumina content in the mixture without impairing refractory properties. The spinel crystals formed from mullite decomposition in the presence of magnesium oxide in fly ash mixtures were less uniform and had cracks, but the resulting properties (MOR, thermal shock resistance) were also improved in mixtures with fly ash levels between 10 and 20 wt.%, with minor impairments to the refractory properties.

The presence of the flux oxides—iron oxides, sodium and potassium oxides (alkalis)—in the raw materials enabled the decomposition of mullite below its melting point, which allowed crystallization of spinel and the creation of the liquid glass phase, promoting sintering in all fly ash mixtures.

In conclusion, the presence of spinel in forsterite ceramics improved the microstructure, the MOR and thermal shock resistance while retaining good refractory properties. Alumina can be substituted by fly ash as a raw material between 10 and 20 wt.% of fly ash (or equal quantities of aluminum oxide) and can achieve similar positive results in the synthesis of forsterite–spinel refractory ceramics.

**Author Contributions:** Conceptualization, M.N. and R.S.; methodology, M.N.; software, M.N.; validation, M.N. and R.S.; formal analysis, M.N.; investigation, M.N.; resources, M.N.; data curation, M.N.; writing—original draft preparation, M.N.; writing—review and editing, R.S.; visualization, M.N.; supervision, R.S.; project administration, M.N.; funding acquisition, M.N. All authors have read and agreed to the published version of the manuscript.

**Funding:** This research was funded by Czech Science Foundation GAČR, grant number 18-02815S, with project name: Elimination of sulphur oxide emission during the firing of ceramic bodies based on fly ashes of class C, and the Internal Grant Agency of Brno University of Technology, specific junior research, grant number FAST-J-20-6196, with project name: Development of refractory materials based on forsterite–spinel–magnesite with regard to corrosion resistance by various corrosive media.

**Conflicts of Interest:** The authors declare no conflict of interest.

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
