# Peer review of "Impact of Fly Ash as a Raw Material on the Properties of Refractory Forsterite–Spinel Ceramics"

_minerals, doi:10.3390/min10090835_

Round 1
Reviewer 1 Report
47-spinel termal expansion is not so low (similar of alumina)
-63 change the sentense: Table 1 présents the chem. comp.obtaied by XRF....
Start "Results"with XRD analysis of raw materials (figure 3)
116-121 write chemical formula of mineral compounds in bracket
124-128:upper : maybe to 2.1 to improve the explanation of compositions
129-139: comment evolution of F series, evolution of R series, compare F and R series,
134-135 introduce figure 6 here because it's the same discussion (only one discussion for RUL)
136-139 upper in desciption of the method in "experimental) but introduce here discussion concerning results!!!!!
discussion conerning MOR??
The back ground noise indicated the presence of iron oxide?????
169--180 itis not explain ifthecurve is related to fired materials or expansion change during the first heating????? very different!... explain
183-187 displace upper and improvediscussionwith table 3
199-203 not a t the right place, see table 3 and figure 5, improve discussion
228 less uniform and had cracks: not previously presented and discussed!!!
idem effect of spinel on mechanical properties!
make sure to have a good link between all parameters (composition -properties for exmaple) and explain it
Reviewer 2 Report
The following full-length article being considered for publication by minerals (MDPI) has been reviewed: “Effect of the iron content within the raw materials on the production of refractory forsterite-spinel ceramics” by M. Nguyen and R. Sokolar.
The work presented is dealing with a very interesting topic the recycling of fly-ash into raw material for the production of forsterite-spinel refractories. Refractory material cost may represent a critical budget item in some processes and it is worth studying the possibility of making unexpensive refractories from recycled raw materials.
The manuscript follows a classical layout: introduction, experiment procedure, results and discussion, and conclusion. The text is well written.
I however would like to encourage the authors considering the few following remarks before acceptation:
Title and objective of the study:
As mentioned in the title and at the end of introduction the author intended to study the effect of iron content on the forsterite-spinel composite ceramics formation. However, iron content effect has an indirect effect through the fly-ash content is the different mixtures and the link between the iron and alkali content and the formation of forsterite-spinel composites is not explicit in the study. I suggest the authors to reconsider both the title and the definition of the objective.
Results and discussion section:
This section would deserve to be differently organised following the suggested outline to help the understanding:
X-ray analyses
SEM analyses
Thermal analyses
Thermal dilatometry
Refractoriness
Other properties which are summarized in table 3.
Sub-sections could also help the reading as it is organised in section 2.
Thermodynamic calculations based on Calphad method would give an idea of the expected phases at equilibrium and could allow for the assessment of the “quality” of the conversion into forsterite and spinel.
The results of the thermal analyses (figure 8 & 9) and refractoriness (figure 6) are presented only for fly-ash content of 10 and/or 20%. The authors explain that results are similar for the higher content and we can trust them but presenting all the results would help to look at the results with a critical eye. In particular, it can be expected the amorphous phase increases with the fly-ash content, even it is not really measured in this study, while its role is critical in the refractoriness and the refractoriness under load. In table 3, refractoriness under load (T0.5) decreases with the fly-ash content probably due the amorphous phase. As a result, the refractory material obtained from the fly-ash content of 30% could not suit for some high temperature applications.
Conclusions section:
According to the last comments about the results and discussion section, I’m not convinced that fly-ash content of 30% is still acceptable for any application and replacing alumina with fly-ash may cause problems. Note that corrosion is a big issue in refractory wear while it has not been investigated in this issue. This may lead to a more moderate conclusion.
Reviewer 3 Report
The article presents an interesting and novel way of recycling fly ash, in this case in forsterite–spinel refractory ceramics. The English presentation of the whole paper is adequate. However, with the publication, some minor revisions are necessary.
- P4, L97: Please revise the unit of heating rate (K/min)
Additionally, please focus on the following aspect: you identified very well the role of iron content interacting with the fly ash and alumina resulting in spinel formation. But in my opinion, you should change the title of your manuscript. There is a great weakness in your investigations: you intended to find the impact of the iron content of fly ash compared to alumina. But to keep the alumina content at the same level, you increase the amount of CCM85 and olivine, which both have high iron content. Consequently, it is hard to state whether the effects are caused by the iron content of the fly ash or the iron content of the MgO-raw materials. With this as solely focus of your manuscript, in combination with the expectations coming with the title, I thought about revising your manuscript. But the novelty of your investigations and results are the role of e.g. the alkalies contained in the fly ash, as well as the decomposition of mullite. Additionally, it is very interesting that you don’t found corundum in the final product. These findings are uncoupled from your – in my opinion - “mistake” of experimental setup increasing the amount of iron-containing MgO-raw materials and merit the publication of your manuscript. That is why I would suggest changing the title, e.g. Impact of fly ash on the properties of refractory forsterite–spinel ceramics. Maybe you will find a more fitting title. If you agree with my suggestion, please check carefully the manuscript for links to the “old” title and intention of investigating the effect of iron. Your manuscript is much more than that!
Round 2
Reviewer 2 Report
The following revised full-length article being considered for publication by minerals (MDPI) has been reviewed: “Effect of the iron content within the raw materials on the production of refractory forsterite-spinel ceramics” by M. Nguyen and R. Sokolar.
The presented work is dealing with the recycling of fly-ash into raw material for the production of forsterite-spinel refractories.
The manuscript follows a classical layout: introduction, experiment procedure, results, discussion, and conclusion. The text is well written English even if it contains some mistakes.
Generally, I am rather happy with the manuscript, which matches the journal topics. The authors answered all the questioned asked by the reviewer and almost all the remarks or suggestions have been taken into account in this revised manuscript. The authors refused to present thermodynamic calculations because they aim at presented this part in a separate on-going paper. Even if I’m not convinced, this argument is defendable. As a result I agree with ist publication providing some minor corrections in the revised text:
Minor corrections:
Figure 2, caption: magnification = 2,000x. Please remove the comma.
Figure 4: Idem
Line 89: “untreated olivine, alumina and fly ash.” Please add a comma after alumina: “untreated olivine, alumina, and fly ash.”
Line 112: “under load and 230×64×54 mm (dimensions…)”: Please add “prism”: under load and 230×64×54 mm prisms (dimensions…)”
Line 116: “4°K” Please remove °: “4 K”
Line 129: The sentence is not clear. Is there a missing verb?
Line 160: “[…] (MgO ) and monticellite[…]” Please add a comma after (MgO): “[…] (MgO ), and monticellite[…]”
